# Long-Term Effect of Hyperbaric Oxygen Therapy on Gait and Functional Balance Skills in Cerebral Palsy Children—A Randomized Clinical Trial

**DOI:** 10.3390/children10020394

**Published:** 2023-02-17

**Authors:** Mohamed E. Khalil, Mohamed A. Abdel Ghafar, Osama R. Abdelraouf, Mariam E. Mohamed, Eman M. Harraz, Reem S. Dawood, Reham A. A. Abouelkheir

**Affiliations:** 1Department of Physical Therapy, College of Medical Rehabilitation, Qassim University, Buraydah 52571, Saudi Arabia; 2Physical Therapy Program, Batterjee Medical College, Jeddah 21442, Saudi Arabia; 3Department of Biomechanics, Faculty of Physical Therapy, Cairo University, Giza 12613, Egypt; 4Department of Physical Therapy for Cardiovacular/Respiratory Disorders and Geriatrics, Faculty of Physical Therapy, Cairo University, Giza 12613, Egypt; 5Department of Physical Medicine, Rheumatology and Rehabilitation, Faculty of Medicine, Mansoura University, Dakahlia 35516, Egypt

**Keywords:** hyperbaric oxygen therapy, gait parameters, functional balance, cerebral palsy

## Abstract

This study aimed to explore the long-term effects of hyperbaric oxygen therapy on spatiotemporal gait parameters and functional balance in children with cerebral palsy. Thirty-nine children with hemiplegic cerebral palsy were randomly allocated to one of two groups: control or study. The children in both groups received traditional physical therapy three times per week for six months. In addition, the children in the study group received hyperbaric oxygen therapy five times/week for eight weeks. The GAITRite system and pediatric balance scale were used to assess spatiotemporal gait parameters and functional balance at baseline, post-intervention, and six months after the cessation of hyperbaric oxygen therapy. Post-intervention means of all measured parameters were significantly higher than pre-intervention means, but only for the study group (*p* < 0.05). However, both groups’ means at the six-month follow up were significantly greater than those at pre-intervention (*p* < 0.05). At the post-intervention and follow-up evaluations, comparisons between groups revealed a statistically significant difference in all measured parameters for the study group against the control group (*p* < 0.05). It can be concluded that adding hyperbaric oxygen therapy to physical therapy rehabilitation could be effective in improving spatiotemporal gait parameters and functional balance in children with cerebral palsy.

## 1. Introduction

Cerebral palsy (CP) is a collection of sensory and motor disorders, as well as postural disorders, caused by non-progressive injury to the immature brain [1]. These obvious motor difficulties are frequently accompanied by cognitive disturbances and other neurologic difficulties [2]. The majority of cases (70–80%) are prenatal in nature; perinatal etiology accounts for 10–20%, and the remainder are of early postnatal etiology [3].

Children with CP have a compromised ability to respond to a loss of balance, in addition to spasticity, muscular weakness, and adaptive muscle length changes. The primary dysfunctions are caused by motor disorders during posture and movement, which limit activities such as walking [4]. Children with CP frequently have abnormal gaits due to motor weakness and a lack of voluntary motor control. Moreover, the gait of children with CP is distinguished by their slow walking speed, a short stride length, and prolonged double support time [5].

Postural control deficits are a primary challenge to motor improvement in CP children. Postural instability limits these children’s ability to complete static and dynamic tasks such as sitting, standing, and walking [6]. Impaired postural control in CP is produced by changes in sensory processing, motor dysfunction, and biomechanical alignment, all of which lead to changes in neuromuscular responses [7].

The major postural dysfunction in CP children is the failure to coordinate the correct sequence of activation of postural muscles, particularly during functional activities and participation limitation [8]. As a result, it is critical to evaluate these children’s functional balance. A wide variety of evaluation processes are available, ranging from accurate equipment to observational evaluations using both qualitative and quantitative procedures [9].

Because of changes in neurological organization, the clinical condition of CP can vary during maturation and its symptoms can fade over time. Occupational, speech, and physical therapy, and palliative care, are the commonly used interventions to improve function and develop compensatory strategies [10]. In recent years, several centers around the world have used hyperbaric oxygen (HBO2) therapy to treat children with CP. 

HBO2 therapy is a medical treatment that entails the delivery of 100 percent oxygen at pressures higher than the surrounding atmosphere (greater than sea level). This allows for a much higher partial pressure of oxygen to be delivered to the tissues, which helps to alleviate hypoxia at the cellular level [11,12]. Furthermore, the therapeutic basis can be interpreted in three ways: physically (hyperbaric 100% oxygen), physiologically (hyperoxia and hyperoxemia), and cellularly [13].

Several mechanisms of cellular and vascular healing mediated by HBO2 have been proposed besides tissue oxygenation. Improved mitochondrial function and cellular metabolism, a better blood–brain barrier and inflammatory responses, less apoptosis, lower amounts of oxidative stress, higher levels of neurotrophins and nitric oxide, and the activation of axonal guiding agents are among these benefits [14,15]. 

Some disorders, including gas gangrene, hypoxia, diabetic foot, carbon monoxide poisoning, stroke, and some thermal burns, have been successfully treated with HBO2 therapy [16]. Moreover, previous studies showed that HBO2 therapy is effective in improving the physical and psychological conditions of CP such as spasticity, gross motor functions, fine motor functions, cognition, auditory attention, and visual working memory [17,18].

Although multiple previous experimental studies on the effects of HBO2 therapy in children diagnosed with CP have been carried out, the proof of the therapeutic effect of HBO2 therapy remains scarce and the results remain inconsistent [19]. One randomized trial found improvements in motor abilities after HBO2, as measured by parent reports [20], while another trial found similar improvements in both the HBO2 and the control groups [21]. Thus, the aim of this study was to explore the long-term effects of HBO2 therapy on spatiotemporal gait parameters and functional balance in children diagnosed with hemiplegic CP.

## 2. Materials and Methods

### 2.1. Study Design

This was a randomized clinical experiment. It was approved by Batterjee Medical College’s ethics and research committee (RES2020-0030) in conformity with the Declaration of Helsinki’s ethical standards. The study was registered at clinicaltrials.gov (NCT05136716).

### 2.2. Participants

Initially, 48 children diagnosed with spastic hemiplegic CP, referred by local pediatric physicians ’offices, were assessed for eligibility. A total of 39 children (24 boys, 15 girls) were chosen based on the following inclusion criteria: age range from 5 to 10 years, being medically cleared by their doctor to use a hyperbaric oxygen hood or mask, the ability to swallow or blow via a straw on command in order to reduce barotrauma, hypertonia range from 1 to 1+ according to the Modified Ashworth Scale, level I or II on the Gross Motor Function Classification System (GMFCS), and the capability to understand and follow instructions. Previous HBO2 treatments, botulinum toxin A treatments, thoracic surgery within six months, significant changes in spasticity medication within the last three months, unstable epilepsy, pulmonary dysfunction, and cardiovascular disease were all exclusion criteria [22]. All children were sorted by sex before being randomly assigned to control and study groups using sealed envelopes. The legal guardian of the child provided written informed consent. Figure 1 displays the flow of participants through the stages of the study.

### 2.3. Sample Size

The sample size was calculated to be 36 children using G*Power (Universities, Dusseldorf, Germany) with an alpha of 0.05, power of 80%, and an effect size of 0.48. 

### 2.4. Procedures

#### Evaluation Procedures

I. Spatio-temporal Gait Parameters.

The GAITRite System (CIR Industries, Clifton, NJ, USA) was utilized to test the temporal and spatial gait parameters. The GAITRite is a 17 ft. long, 2.9 ft. wide instrumented carpet. The active part of the walkway is 14 ft. long and 2 ft. wide with pressure sensors implanted in the horizontal grid of the carpet (each divided by 0.50 inch). The sensors recorded the location of the activated sensors and the time of sensor activation/deactivation with each step, allowing for dynamic pressure mapping during walking and the computation of a variety of temporal and distance gait parameters to be recorded. The sensors provided information about the two-dimensional geometry of the footprints as the child moved across the carpet. The walkway was connected to a computer, and the GAITRite Version 3.2b software package was used to measure and record spatiotemporal parameters [23]. This tool has demonstrated an outstanding test–retest reliability (ICC = 0.91) [24].

Before the actual measurement, all subjects were given a 5 min acclimation period. The child then completed three walking trials at their own pace with their own footwear to collect data. When gait data for three full strides at a steady pace on the walkway were recorded, the trial was rated successful. A meter was installed at both ends of the walkway to allow for acceleration and deceleration stages [25]. Parameters selected for this study were walking velocity, cadence, step length, step-length symmetry ratio, and single limb support.

II. Functional Balance.

To assess the functional balance of the children, the pediatric balance scale (PBS) was used [26]. Collected PBS scores vary from 0 to 56 for the 14 tasks examined, with higher scores suggesting higher postural control. The items are graded on a five-point scale, with 0 indicating unable to complete the activity without assistance and 4 indicating complete independence. The time it takes to maintain a position, the distance the upper limb may reach in front of the torso, and the time it takes to accomplish the activity all contribute to the score. The test was carried out while the child was dressed and wearing their usual brace and/or gait-assistance device [27,28]. All types of PBS reliability were assessed by Her et al. [29] and reported to be high in children with CP.

Functional balance and spatiotemporal gait parameters were measured three times: pre- and post-intervention, and six months after the stopping of HBO2 therapy to examine the long-term impact.

### 2.5. Intervention Procedures

The children in the study group were treated with HBO2 therapy and received 100% oxygen at a pressure of 1.7 atmospheres (atm) in a multi-patient hyperbaric chamber. A 15 min period was used for the gradual increase in the chamber pressure. When 1.7 atm was reached, the participants were fitted with acrylic hoods or a mask and 100% O2 was delivered. For 60 min, the participants breathed 100% O2. Afterwards, participants breathed air for 15 min to decompress. Each session took approximately 90 min to complete. The treatments were given 5 times/week for eight weeks, for a total of 40 treatments [18]. Prior to the treatments, the participants were checked by a physician who specialized in hyperbaric medicine to reduce the risk of adverse effects.

One hour of traditional physical therapy was provided to both groups three times per week for 6 months. The neurodevelopmental treatment approach included tight muscle stretching, weak muscle strengthening, postural reaction training, proprioceptive training, and walking training.

### 2.6. Statistical Analysis

The authors used the Statistical Package for Social Sciences (SPSS) for Windows, version 20.0 (Armonk, NY, USA: IBM Corp.). The choice of parametric statistical methods in this study was based on the results Shapiro–Wilk test that ensured the normal distribution of the collected data. An unpaired *t*-test was used to investigate the significance difference between the mean of both groups regarding age, weight, and height, while the χ^2^ test was used to check the difference between groups in terms of sex, hypertonia, and GMFCS level. Finally, Multivariate analysis of variance (MANOVA) was used to compare the means of the study and control groups in terms of all the study’s outcome measures. A post hoc test with the Bonferroni corrections was used in the case of significant results. With 95% confidence intervals, a probability of *p* < 0.05 was statistically significant.

## 3. Results

The independent *t*-test showed no significance difference between the control and the study group at baseline in terms of age, height, and weight (*p* = 0.74, 0.69, and 0.47, respectively). Additionally, the chi-square test showed that the difference of the means of both groups in terms of sex, hypertonia, and GMFCS level were not significant (*p* = 0.81, 0.76, and 0.93, respectively) as illustrated in Table 1.

Within-group comparisons using MANOVA showed that the post-intervention means of PBS, walking velocity, cadence, step length, step-length symmetry ratio, and single limb support were significantly higher than the pre-intervention means for the study group (*p* = 0.007, 0.009, 0.021, 0.009, 0.011, and 0.008, respectively) with a non-significant difference for the control group (*p* = 0.658, 0.726, 0.863, 0.785, 0.732, and 0.947, respectively). Moreover, the means at the six-month follow up were significantly higher than those at pre-intervention for the study group (*p* = 0.003, 0.006, 0.011, 0.006, 0.006, and 0.003, respectively) and the control group (*p* = 0.012, 0.018, 0.005, 0.019, 0.019, and 0.009, respectively) as illustrated in Table 2. Between-group comparisons showed no statistically significant difference between the pre-intervention means of all outcome measures in both groups (*p* > 0.05). However, comparisons between variance (mean differences) in the pre–post and pre-follow-up evaluations of both groups showed statistically significant differences in all measured parameters in the study group compared to the control group (*p* < 0.05). 

## 4. Discussion

The present study was conducted to examine HBO2’s effects on gait and balance in spastic CP. The results of the current study showed a significant improvement in gait parameters and functional balance measured after eight weeks and then after six months (long term) in the children who were treated with HBO2 therapy.

These findings could be attributed to an increased oxygen availability to neural cells, which may stimulate them to function normally by reactivating them metabolically or electrically, resulting in angiogenesis and other signs of healing [30]. Furthermore, HBO2 therapy may induce the proliferation of endogenous neural stem cells, contributing to the repair of the injured brain. Increased oxygen to brain tissues is the basic denominator underlying all of these mechanisms [31]. HBO2 may simply supply the missing oxygen required for these regeneration processes, allowing metabolic change to occur [32,33].

HBO2 therapy advantages in CP are related to a rise in dissolved oxygen in plasma and tissue, which helps regeneration in light of the theory that increased oxygen availability to injured brain cells can reawaken normal functions [34]. Based on the results of the present study, HBO2 can be recommended as an adjunctive therapy to conventional rehabilitation programs to reduce spasticity and improve gross motor development, which is in line with previous research findings on HBO2 therapy in CP [18,21].

A randomized controlled study conducted by Collet et al. [21] reported significant improvements in gross motor functions, disability score, and other outcomes after 40 sessions of HPO2. Moreover, Mukherjee et al. [18] concluded that HBO2 therapy can enhance the outcomes of standard rehabilitation and motor function. The authors reported a significant improvement in GMFM scores in the group who was treated with hyperbaric therapy. Similar results were reported by Azhar et al. [35] who found a significant increase in the GMFM, ADL score, and, to a smaller extent, hearing and speaking after a two-month treatment with BBOT. Additionally, the outcomes of the present study came in parallel with Montgomery et al. [36] in the matter of children diagnosed with CP being greatly improved due to a reduction in spasticity, improved GMFM in three of the five items in the gross motor functional measurement test, and improved fine motor function in three of the six hand tests.

The improvement in gait parameters and functional balance in the study group after 6 months could be explained by the fact that HBO2 therapy enhances nerve cell regeneration and re-growth, starting about 4–6 months after beginning therapy. Physical therapy treatment given during this period magnifies the rate of neurodevelopmental progress and significantly closes the gap between recognized growth patterns and observed growth rate [37].

These results are in agreement with Machado [38] who found that patients experienced persistent reductions in spasticity and better motor control in a follow-up assessment six months after the cessation of HBO2. In addition, the parents claimed that their child had improved in terms of balance, attentiveness, and intelligence, as well as having fewer convulsions and episodes of bronchitis. Heuser and Uszler [39] emphasized the continued beneficial effects of HBO2 beyond the treatment endpoint. Based on the results of their study, the functional improvements achieved after a long course of HBO2 were retained without further treatment for at least three years. This may be due to re-canalization of atrophied vessels or the in-growth of neo-vasculature.

On the other hand, our results disagreed with those of Lacey et al. [22], which found no progress in gross motor skills of spastic CP children who received 40 sessions of HBO2 therapy. This might be because the mentioned trial was terminated prematurely after two months, preventing the results from reaching a statistically significant level. It might also be because they used a different age group (3–8 years), used HBO2 therapy with a pressure of 1.5 atm, and used selected assessment tools that were different to ours. Finally, they allowed all recognized clinical categories of spastic CP.

Moreover, the current study disagrees with Jain, [40] who stated that there is no effect of HBO2 therapy on spasticity or gross motor skills. This could be explained by the inadequate number of treatments in Jain’s study; based on his results, the author recommended 40 treatments at 1.7 atm/60 min to produce the gains. Additionally, the findings of this study contradicted those of Katanani [41], who reported in a pilot study that while the results appeared to be encouraging, no statistically significant value was found in the gross motor skills, sociality, and awareness of CP children who received 40 sessions of HBO2 therapy. As a result, the advantages of HBO2 therapy in CP children are debatable, which might be attributable to the subjective instrument utilized, such as a questionnaire filled out by parents or care providers, the lack of a control group, and the small number of CP children and children with intellectual disabilities who participated in the study.

Additionally, the present study found significantly higher improvements in all measured outcomes after 6 months in the study group compared with the control group, which contradicted Hegazy et al. [42], who stated that the results showed no statistically significant difference between the control and the HBO2 therapy groups. This could be attributed to the difference in the children’s age (4–7 years), the duration of 3 months, the number of treatment sessions (60 sessions), and assessment tools, as they used electromyography (EMG) to assess spasticity through the H/M ratio and the Box and Block Test (BBT).

The limitations of this study are as follows: First, we did not assess the anatomical and physiological changes in the brain after application of HPO2 therapy by using a single-photon emission computerized tomography (SPECT) scan. Second, a double- or single-blinded experimental model was not used, since the same group of researchers conducted both the HBOT and the assessment processes. This entails a considerable risk of observer bias. Third, a medium effect size was chosen to detect decent changes in the means of the outcome measures. More large-scale randomized controlled studies are required to determine who the best candidates are and what the optimum HBO2 strategy is for children with CP.

## 5. Conclusions

The current study findings indicate a significant improvement in gait parameters and functional balance immediately post-intervention and in a follow up after 6 months. Based on the results of this study, HPO2 therapy when added to physical therapy rehabilitation could be an effective intervention for improving spatiotemporal gait parameters and functional balance over time in children diagnosed with spastic CP. Moreover, the changes in function may extend over a prolonged period of time, even after the cessation of HBO2 therapy.

## Figures and Tables

**Figure 1 children-10-00394-f001:**
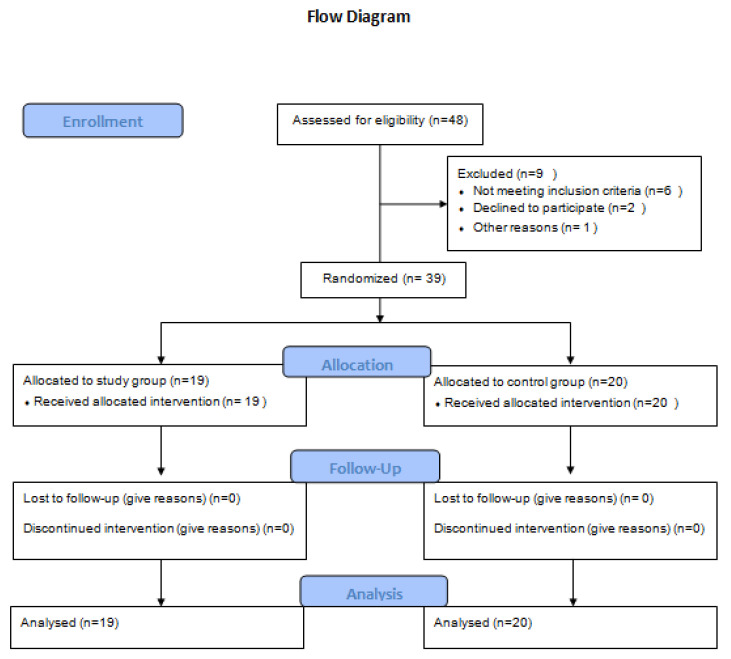
The flow diagram of the of the participants through the study.

**Table 1 children-10-00394-t001:** This is a table of the demographic characteristics of children in both groups.

Groups	Study Group *n* = 19	Control Group *n* = 20	*p*-Value
	Mean ± SD	Mean ± SD	
Age (years)	7.31 ± 2.29	8.52 ± 1.48	0.74 ^a^
Height (cm)	130.45 ± 5.08	128.95 ± 6.71	0.69 ^a^
Weight (kg)	31.99 ± 6.83	33.54 ± 4.36	0.47 ^a^
Sex, Boys/Girls	(13/6)	(13/7)	0.81 ^b^
Hypertonia, 1/1+	(3/16)	(4/16)	0.76 ^b^
GMFCs, Level, I/II	(5/14)	(6/14)	0.93 ^b^

Data are illustrated as mean ± SD standard deviation, ^a^ refers to independent *t*-test, ^b^ refers to chi-square test, *p*-Value is significant at >0.05.

**Table 2 children-10-00394-t002:** This is a table of pre-intervention, post-intervention and follow up of all outcome measures of the study and control groups.

Variables	Groups	Pre	Post	Follow Up	Pre–Post-Intervention Comparison		Pre-Follow-Up Intervention Comparison	
		Mean ± SD	Mean ± SD	Mean ± SD	*p*-Value	MD (95% CI)	*p*-Value	MD (95% CI)
PBS	Study group n = 19	36.81 ± 6.99	44.57 ± 8.46	49.67 ± 9.03	0.007 *	7.76 (2.65–12.87)	0.003 *	12.86 (11.77–13.94)
	Control group n = 20	38.62 ± 7.73	39.03 ± 7.81	44.05 ± 8.28	0.658	0.41 (0.35–0.44)	0.012 *	5.43 (4.17–6.69)
Walking velocity (m/s)	Study group n = 19	0.3 ± 0.19	0.39 ± 0.13	0.45 ± 0.15	0.009 *	0.09 (0.07–0.11)	0.006 *	0.15 (0.13–0.18)
	Control group n = 20	0.31 ± 0.2	0.33 ± 0.09	0.39 ± 0.15	0.726	0.02 (0.015–0.034)	0.018 *	0.08 (0.062–0.097)
Cadence step/min	Study group n = 19	97.71 ± 2.62	110.82 ± 3.78	120.31 ± 4.56	0.021 *	13.11 (10.97–15.25)	0.011 *	22.6 (20.54–24.65)
	Control group n = 20	95.33 ± 2.96	97.48 ± 3.15	106.53 ± 4.12	0.863	2.15 (0.19–4.11)	0.005 *	11.02 (9.88–12.15)
Step length (m)	Study group n = 19	0.26 ± 0.15	0.35 ± 0.11	0.41 ± 0.07	0.009 *	0.09 (0.003–0.18)	0.006 *	0.15 (0.12–0.17)
	Control group n = 20	0.28 ± 0.09	0.30 ± 0.13	0.35 ± 0.17	0.785	0.02 (0.014–0.03)	0.019 *	0.07 (0.059–0.080)
Step-length symmetry ration	Study group n = 19	0.71 ± 0.13	0.77 ± 0.29	0.86 ± 0.3	0.011 *	0.06 (0.03–0.08)	0.006 *	0.15 (0.13–0.16)
	Control group n = 20	0.69 ± 0.17	0.71 ± 0.18	0.76 ± 0.21	0.732	0.02 (0.092–0.13)	0.019 *	0.07 (0.055–0.084)
Single limb support (% GC)	Study group n = 19	34.29 ± 8.11	40.72 ± 6.5	46.93 ± 8.47	0.008 *	6.43 (1.59–11.27)	0.003 *	12.64 (10.08–15.20)
	Control group n = 20	32.64 ± 6.73	34.43 ± 4.68	39.14 ± 6.88	0.947	2.79 (2.54–3.87)	0.009 *	6.5 (5.72–7.27)

Data are illustrated as mean ± SD standard deviation, n number of participants, PBS pediatric balance scale, GC gait cycle; MD mean difference, CI confidence interval, * *p* value is significant at <0.05.

## Data Availability

The materials that support this manuscript are available from the corresponding author upon reasonable request.

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
