# Peer review of "Long-Term Effect of Hyperbaric Oxygen Therapy on Gait and Functional Balance Skills in Cerebral Palsy Children—A Randomized Clinical Trial"

_children, 2023, doi:10.3390/children10020394_

Round 1

Reviewer 1 Report

Great study!!  I hope you continue to pursue this research with larger groups and more measurements.  What a great opportunity for those persons diagnosed with CP. Great work!

My changes below are basically editorial for clarity and consistency.

Abstract

2d line and last line – consider title “children diagnosed with cerebral palsy and in the last line it should state balance in children diagnosed with spastic CP. Add abbreviation as it was identified a the top.

4th line ….change the number 6 to the word six for consistency with the remainder of the abstract 

Body of manuscript:

Consider changing  all references of “children with CP” to “children diagnosed with CP”

Page 2 line 57 and 58 state that HBO2 has been used to treat children with CP. Could you add a little about what conditions have been treated successfully with HBO2?

Line 80 on page 2 under materials and methods uses the term ‘spastic children with hemiplegic cerebral palsy’. Please state 48 children diagnosed with spastic CP with hemiplegia.  Also change the cerebral palsy to CP. Please be consistent with abbreviations. 

Also same paragraph line 84 uses the term ‘medically free’. What does that mean?

In the same paragraph the term HBO is used instead of HBO2. Is there a difference? If not make sure that all abbreviations are consistent

Also be consistent with with use of numbers. In the first line of the participants paragraph you spell out forty eight but the number  6  is used for 6 months. Please be consistent. It is good to use the number 48.

Page 3 , line 109-110. It is stated that they used ‘their own footwear’. Can you describe the footwear? Was it closed toed shoes, slip ons, sandals or barefoot?

 Page 3 line 123. Please use the term children diagnosed with CP

 Numerous times throughout the manuscript you switch to writing numbers out.  Please check all and be consistent. Using actual numbers is good. 

In the Intervention Procedures paragraph, you state that a multi patient hyperbaric chamber was used. How many children were in one chamber? Also what does ‘atm ‘ stand for? Please clarify.

 Is it a hyperbaric physician? Or a physician who specializes in hyperbaric treatment?

Did the participants in the control group receive any kind of sham HBO2 tx?  Or just the traditional PT? and were they offered HBO2 treatments after the study was completed? 

Under results, page 4 line 152, it should say t-test ‘showed’ instead of ‘should’ 

Under the Discussion, the term hyperbaric oxygen was spelled out instead of using the abbreviation HBO2, please be consistent. Also please check throughout for consistency in the term children “diagnosed” with spastic CP  instead of children with spastic CP 

Page 5 line 200, the term after does not need to be capitalized. Also there is a couple extra spaces between the words Moreover, and Mukherjee.

Page 5 , what does GMFM stand for? Please clarify.

There appears to be some inconsistencies in the references formatting. Please check carefully.

Author Response

The authors thank the reviewer for his positive comments and careful review, which helped improve the manuscript.

Reviewer 2 Report

Overall, this is a well-written manuscript reporting results of random control comparing control group (traditional therapy) to study group (traditional therapy plus hyperbaric oxygen therapy). 

I believe there are several clarifications/modifications that are required before accepting manuscript for publication. Please see attached word document.

Author Response

(The authors gave the same response as above.)

Reviewer 3 Report

Dear Authors,

First of all, I would like to congratulate you for the research carried out. It is a well-developed and quality work. However, the manuscript has some formal errors and limitations that should be addressed before its possible publication in this Journal.

ABSTRACT:
Please remove abbreviations from this section.
If you decide to report the p-values obtained, please do so in a faithful and concrete manner. Not simply indicating whether the result was greater or less than 0.05.

INTRODUCTION:
This section could be complemented by explaining how complex the assessment of postural control in children and especially in children with neurological pathology can be (doi: 10.1016/j.gaitpost.2021.04.027 // 10.3390/diagnostics11010008).

METHODOLOGY:
The prior sample size calculation should be complemented by calculating the power of the final sample analyzed. In addition, since an effect size of 0.48 was previously established it should be included as a limitation of this research.

STATISTICAL ANALYSIS:
The applied tests should be complemented with the calculation of their effect sizes.

RESULTS:
To transmit descriptive data results (means and standard deviations) it is sufficient to include only one decimal place.
In addition, zeros as the last decimal place mean nothing.
Tables should be thoroughly revised so that they do not repeat information in a useless way. In addition, they should be considered pieces of information independent of the text and, therefore, any abbreviations or acronyms should be explained at the bottom (if it is really necessary to include them).

DISCUSSION:
This section is correct and appropriate.
However, it could also benefit from the inclusion of new references (such as those recommended for the Introduction).
In addition, limitations should be honestly acknowledged and expanded upon.

CONCLUSIONS:
Although correct, they should be expanded with the interpretation and clinical impact of the results obtained.

Kind regards

Author Response

(The authors gave the same response as above.)

Round 2

Reviewer 2 Report

Thank you for addressing my questions and comments in a very thorough manner.

Author Response

(The authors gave the same response as above.)

Reviewer 3 Report

Dear Authors,

First of all, I would like to congratulate you on your work and efforts to improve the manuscript in response to the reviewers' corrections.

Personally, I now consider that the manuscript can be accepted for publication in this Journal.

Kind regards

Author Response

(The authors gave the same response as above.)
